# Generation of Photon Orbital Angular Momentum and Its Application in Space Division Multiplexing

**Temitope M. Olaleye \*** , **Paulo A. Ribeiro \*** and **Maria Raposo**

Laboratory of Instrumentation, Biomedical Engineering and Radiation Physics (LIBPhys-UNL), Department of Physics, NOVA School of Science and Technology, Universidade NOVA de Lisboa, 2829-516 Caparica, Portugal; mfr@fct.unl.pt

\* Correspondence: o.mary@campus.fct.unl.pt (T.M.O.); pfr@fct.unl.pt (P.A.R.)

**Abstract:** In the last three decades, light's orbital angular momentum (OAM) has been of great interest because it has unique characteristics that make it sought after in many research fields, especially in optical communications. To address the exponentially increasing demands for higher data rates and capacity in optical communication systems, OAM has emerged as an additional degree of freedom for multiplexing and transmitting multiple independent data streams within a single spatial mode using the spatial division multiplexing (SDM) technology. Innumerable research findings have proven to scale up the channel capacity of communication links by a very high order of magnitude, allowing it to circumvent the reaching of optical fiber's non-linear Shannon limit. This review paper provides a background and overview of OAM beams, covering the fundamental concepts, the various OAM generators, and the recent experimental and commercial applications of the OAM-SDM multiplexing technique in optical communications.

**Keywords:** photons; orbital angular momentum; spatial division multiplexing; optical communication

## 1. Introduction

The technological boom over the past few decades has resulted in the exponential growth of global internet users to over five billion as of the second quarter of the year 2022 [1]. This number has already matched Cisco's projection for 2023 [2]. This surge has inherently caused the need for better ways to improve the channel capacity of the various communication links used by the devices of internet users such as the usage of high-bandwidth applications [3], high-definition video conferencing contributed by remote working, an increasing number of heavily used social media apps, distributed computing systems, sensor networks, and cloud-based services to name a few. This has unabatedly resulted in the quest to find a solution to the exponential traffic growth on communication networks which is generally believed, based on information theory [4] to be approaching the non-linear Shannon limit [5,6]. Charles Kao's [7] 2009 physics Nobel-prize-winning research published that single-mode silica-based optical fibers are the future of communications; however, over the years, its scalability is fast running out—this is commonly called a "capacity crunch", indicating that the capacity limits of the transmission channels are approaching. The implication is that when the current communication channels reach their maximum capacity, more optical cables will be required to be installed to cope with the increasing demand [8]. However, this may not be the best option because upgraded fiber will not even be able to address system capacity scaling issues [9]. Hence, additional ways to multiplex with the existing techniques for a higher channel capacity are required. In the optical communications research community, this problem opened the need to adopt multiplexing techniques such as orbital angular momentum (OAM) and space division multiplexing (SDM) which allows information to be carried simultaneously in multiple modes, theoretically infinite, and which can also be easily adapted with the currently existing technologies.

Recently, Allen et al. [10] discovered that a Laguerre Gaussian (LG) laser mode has twisted phase fronts with a well-defined orbital angular momentum (OAM) of $\pm l\hbar$ per photon where $\pm l$ is the azimuthal mode index also known as OAM mode ranging from $\pm 1$ to infinity, acting as the integral multiple of the reduced Planck's constant $\hbar$. The helical shape and orthogonality of OAM beams are major differentiating characteristics that make it applicable in many unique ways such as in optical tweezers [11,12], atomic manipulation [13–15], nanoscale microscopy [16–19], wireless communication [20], information encoding [21], quantum information processing [22], data storage [23–25], and optical communication, which is the main interest of this review.

The bandwidth/capacity of a communication system is determined by the number of orthogonal modes that are available for information to be encoded. This makes orthogonal beams such as OAM beams the best bet as they can be encoded with information and multiplexed with existing multiplexing techniques for an additional degree of freedom to boost the data capacity of transmission channels. Orbital angular momentum multiplexing is a subset of space division multiplexing (SDM), which sees how separately distinguished modes such as OAM modes can be used as alternatives to other multiplexing techniques. Winzer's report [26] in Figure 1 also shows that SDM technology will be adopted for the desired systems capacities in the next decades. Interestingly, OAM multiplexing has been named as one of the promising technologies for the multi-terabyte per second (Tb/s) and 6G networks [27] and is expected to provide superior performance to the 5G ecosystem.

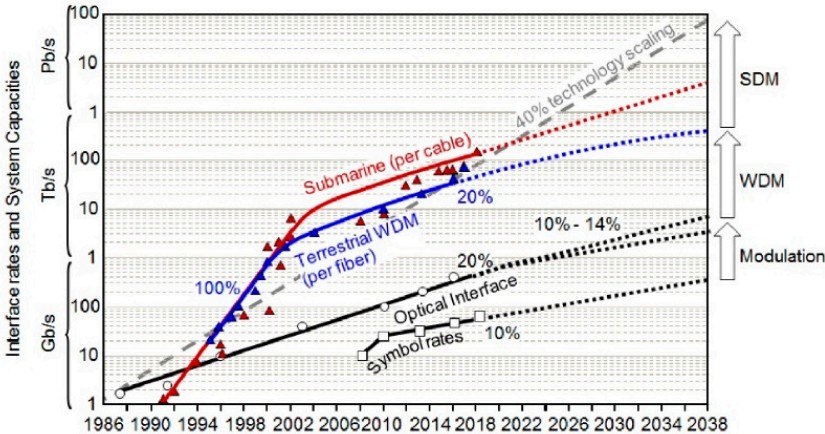

**Figure 1.** The long-term evolution/trend of commercial optical transmission systems over the past 30 years and extrapolations for the next 15 years [26] with the assumption that the technology continues to grow at a 40% scale according to the historic trend [9]. Reprinted from ref. [26].

The ever-increasing demands for higher data capacities and additional degrees of freedom in optical communications have driven the evolution of communication channels across various modulation and multiplexing technologies over the years [28–30]. Multiplexing allows the simultaneous transmission of combined signals through a single channel with the aim of increasing the capacity of the communication system. In optical communication systems, information is encoded in light signals and transmitted through optical fiber or free space. Light signals have several physical dimensions which can be modulated and multiplexed. The dimensions are time, amplitude/phase, wavelength/frequency, polarization, and spatial dimension. Since the origin of multiplexing [31,32], these dimensions have been explored in several multiplexing techniques (shown in Figure 2) to improve the bandwidth of communication channels, including the following: the amplitude/phase dimension for quadrature amplitude modulation QAM [33,34], which transmits information by combining the phase and amplitude of a carrier wave; time for time-division multiplexing TDM, which is one of the earliest multiplexing techniques used in optical communications [35,36] where several independent signals are transmitted through a channel of different timeslots; polarization

for polarization-division multiplexing PDM [37], a multiplexing method which combines two linear and orthogonally polarized signals over the same carrier frequency and doubling the system capacity; the wavelength/frequency for wavelength/frequency-division multiplexing WDM/FDM [38,39] where multiple signals of different wavelength/frequency are transmitted over a single channel; and spatial multiplexing (SM) also known as space-division multiplexing (SDM), termed the next frontier in optical communication [40,41], which uses the multiplicity of spatially separated/unique orthogonal modes (such as OAM modes) to transmit multiple optical data. In multiplexing, the orthogonality of the transmitted signals in the various dimensions is paramount [42]. It ensures that the uniquely separated signals can be multiplexed, transmitted, and demultiplexed without impacting the performance of the transmitted information on each signal.

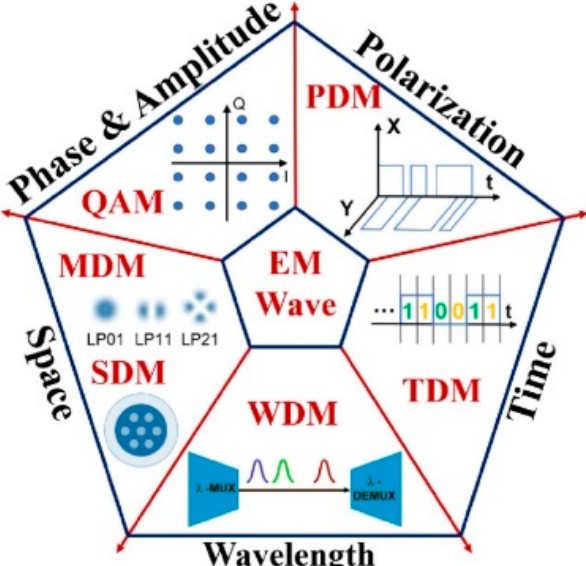

**Figure 2.** Multiplexing and modulation techniques in optical communications. (TDM: time-division multiplexing; WDM: wavelength-division multiplexing; SDM: space-division multiplexing; MDM: space-division multiplexing; QAM: quadrature amplitude modulation; PDM: polarization-division multiplexing. Reprinted from ref. [43].

This review is a valuable resource for individuals new to the OAM field and seeking a concise overview and accessible summary of the vast existing literature. It makes the complex concepts of OAM more understandable to a broader audience and offers an extensive awareness of the potential of OAM as a crucial technology for achieving greater channel capacity in future communication networks.

*Roadmap of the Review*

This review addresses readers who want to have general knowledge about orbital angular momentum and its role in scaling channel capacity and meeting the escalating capacity demands of future communication systems. First, the background of OAM is introduced along with its physical characteristics and connection to the helical phase structure of light, emphasizing their special qualities and potential to boost communication capacity.

Next, the various approaches used to generate OAM modes are addressed. These methods consist of the spatial-generating method and the fiber-generating methods. We also look at their comparisons, advantages, and disadvantages. Furthermore, we discuss how the generated OAM can be detected and how they propagate. The experimental breakthroughs and demonstrations of OAM multiplexing in SDM with other multiplexing techniques such as PDM and WDM are reported. Moreover, a highlight is given about the foundational elements of SDM, such as multicore fibers, and few-mode fibers, and

how they enhance channel capacity, spectral efficiency, and increase data rates. Lastly, the necessity for OAM-compatible devices and the recently commercialized OAM-SDM products and devices are mentioned.

## 2. Background of OAM

According to Maxwell [44], the fundamental property of an electromagnetic wave is energy and momentum. It has a linear momentum $\boldsymbol{p} = m\boldsymbol{v}$ and an angular momentum $\boldsymbol{L} = \boldsymbol{r} \times \boldsymbol{p}$ when the particle moves at a position $\boldsymbol{r}$ from the origin. The difference between angular momentum and linear momentum is that angular momentum, a vector quantity, has a rotation or spin about an axis and a clear direction in which it acts. For an electromagnetic wave, angular momentum in the $z$ direction requires a component of linear momentum in the $x$, $y$ plane, a light beam with a transverse momentum component. The linear momentum density $\boldsymbol{\rho} = \epsilon_0 \boldsymbol{E} \times \boldsymbol{B}$ is related to the angular momentum density $\boldsymbol{j}$ through $\boldsymbol{j} = \boldsymbol{r} \times \boldsymbol{\rho}$ where $\epsilon_0$ is the medium dielectric permittivity, and $\boldsymbol{E}$ and $\boldsymbol{B}$ are the electric and magnetic fields, respectively. Hence, it follows that the angular momentum of an EM wave propagating in the $z$-direction requires a component of the electric and/or magnetic field also moving in the $z$-direction.

By 1936, Beth demonstrated [45] Poynting's theory [46] of the rotation of circularly polarized light beams transmitted through a half-wave plate. He noticed that the wave plate reversed the handiness of the transmitted polarized light, hence also reversing the angular momentum of the light beam. The change in the light's angular momentum resulted in a torque $\boldsymbol{\tau}$, a change in angular momentum with respect to time, which caused the wave plate to rotate. In modern physics where momentum per photon is considered, this change in handedness or spin in the left and right directions is known as spin angular momentum (SAM) with the value $\pm\hbar$, where $\pm$ is the handedness of the circular polarization and $\hbar$ is the Planck's constant, a conserved quantity. Beth believed that a photon should be able to carry a more complex angular momentum which would result in a much greater momentum transfer of $\hbar$ as in the case of circularly polarized light.

### 2.1. OAM in High-Order Transmission

During the high-order transmission experiment, Charles G. Darwin [47] considered a light emitted at a short radius from the atomic orbit and found that the linear momentum of the emitted photon resulted in an extra torque $\boldsymbol{\tau}$ on the atomic orbit. This torque is now called the orbital angular momentum (OAM). Orbital angular momentum arises when beams move in a non-perpendicular direction to the propagation axis. In geometric optics approximation, the light which makes up the OAM beams is skewed. Skewed rays are rays that travel through an optical fiber without passing through the axis of the optical fiber. OAM has $\pm l\hbar$ independent states per photon, $l$ being an integer. The signs indicate the handedness with respect to the beam direction where $-$ signifies the clockwise direction and $+$ signifies the anticlockwise direction.

When a particle rotates about position $\boldsymbol{r}$ it is said to have angular momentum; hence, a particle at a position $\boldsymbol{r}$ revolving/spinning around the orbit of an atom is said to possess an orbital angular momentum $mvr = n(h/2\pi)$ where $m$ is the mass of the particle, $v$ is the velocity, $r$ is the radius of the orbital, $(h/2\pi)$ is the unit of the energy quanta, and $n$ is an integer. Bohr proposed that as an electron absorbs more energy $n$, it gains more orbital angular momentum [48]. The same has been said of a photon, according to Poynting, that must be accompanied by an angular momentum as it transforms from linear polarization to a circularly polarized light.

### 2.2. The Total Angular Momentum of a Photon with OAM

For over 200 years, the spin angular momentum (SAM) property of light which is associated with circular polarization has been extensively investigated. Initial reasoning from Cohen-Tannoudji et al. [49] and Nishijima in 1965 [50] was that it was meaningless to separate the angular momentum of a photon into the spin and orbital parts and [51]

also pointed out that not all light waves can possess orbital and spin angular momentum. In 1992, 30 years ago, four researchers working in a quantum optics lab debunked this when they theoretically identified these two distinct properties in a photon. It was shown in [52] that the spin angular momentum and orbital angular momentum are well-defined and separately measurable quantities of a photon. In theory, the OAM beams have an unlimited degree of freedom and enhance information-carrying capacity in both classical and quantum optical communications.

Hence, it is now generally accepted that the total angular momentum $J$ of a photon consists of both SAM, the intrinsic component of the photon which is associated with wave polarization, and OAM, the extrinsic component that represents the spatial distribution of the electromagnetic wave/photon:

$$J = \text{SAM} + \text{OAM} \tag{1}$$

With SAM and OAM defined, respectively, by:

$$\text{SAM} = \varepsilon_0 \int (\boldsymbol{E} \times \boldsymbol{B}) d^3 \boldsymbol{r} \tag{2}$$

$$\text{OAM} = \varepsilon_0 / 2iw \sum_{i=x,y,z} \int \boldsymbol{E}^{i*} (\mathbf{r} \times \nabla) \boldsymbol{E}^i d^3 \tag{3}$$

$\boldsymbol{E}$ and $\boldsymbol{B}$ are the electric field and the magnetic field, respectively, $\varepsilon_0$ is the vacuum permittivity, and $w$ is the speed of light. The i-superscripted symbols denote the Cartesian components of the corresponding vectors. The conservation of total angular momentum $J$ assumes that without the presence of external forces, no energy of the wave is lost as it propagates. The OAM property of a photon can be multiplexed with the polarization property (SAM) of the same photon, which is independent of OAM, to increase channel capacity. Table 1 summarizes the properties of SAM and OAM.

**Table 1.** Overview of SAM and OAM features. $\sigma$ represents the number of states the circularly polarized waves have whereas $l$ is the azimuthal mode index/OAM mode ranging from 1 to infinity $\infty$. It indicates the number of independent twists ($2\pi$ phase shift in the azimuthal direction) that a beam possesses. $\pm$ indicates the handedness of the beams; $-$ signifies that that beam is propagating in a clockwise direction and + signifies the anticlockwise direction.

| Angular Momentum | SAM | OAM |
|---|---|---|
| Wavefront polarization | Circularly polarized waves | Helically phased waves |
| Angular momentum per photon | $\Sigma\ \sigma h$ | $\Sigma\ l\hbar$ |
| Photon state number | $\sigma = \pm 1$ | $l = \pm(1, 2.3, \ldots, \infty)$ |

### 2.3. Laguerre-Gaussian Modes

Twisted light beams are a new type of laser beam that have separate OAM modes, also known as Laguerre-Gaussian (LG) modes. Laguerre-Gaussian modes are the most common form of a helically phased beam. Each mode is differentiated by the number of spirals or twists it possesses as in Figure 3.

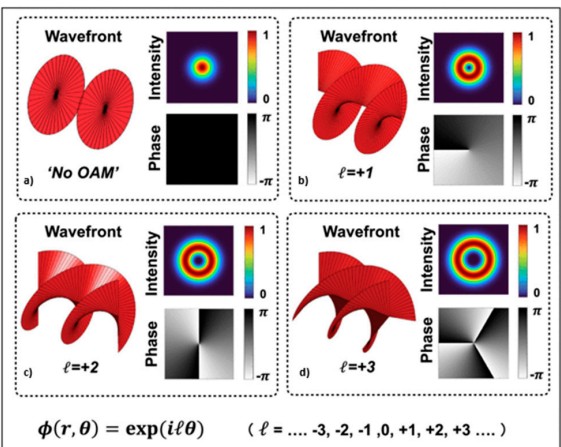

**Figure 3.** Wavefront, phase profile, and intensity profiles of OAM beams. (**a**) *l* = 0. This is a Gaussian beam with a donut-shaped intensity profile and no twist/OAM in the wavefront; (**b**) *l* = 1, which has one twist per wavelength; (**c**) *l* = 2, which has two twists per wavelength; (**d**) *l* = 3, which has three twists per wavelength. Helical beams (**b**–**d**) are all orthogonal to each other with each traveling in the anticlockwise direction indicated by the + sign. These beams can carry independent information on the same transmission and channel frequency. Reprinted from ref. [53].

Allen et al. stated that the amplitude of a Laguerre-Gaussian mode has an azimuthal dependence of $e^{il\varphi}$ and that the Laguerre polynomial distributions of amplitude $TEM_{pl}$ possess well-defined orbital angular momenta where p is the radial dependence and l is the azimuthal dependence. When $p = 0$ and $l \neq 0$, the LG beam is called $TEM_{01}$ which is more applicable in micromanipulation [54]. The intensity profile of a Laguerre-Gaussian beam $LG_{lp}$ varies with *p* and *l*. When *p* and *l* are zero (0), one has a Gaussian intensity profile, a spot, but as the *l* value increases or decreases, the intensity profiles are seen as an optical vortex indicated by the dark centers surrounded by bright annular rings of helical beams. The higher the *p*-value, the more rings or intensity profiles as seen in Figure 4.

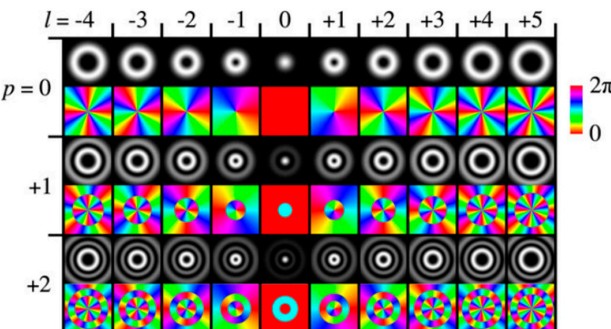

**Figure 4.** Intensities (the rings) and phase profiles of LG modes. In each row, the azimuthal index *l* increases from left to right whereas the radial index *p* increases from top to bottom. When *p* = 0 and *l* = 0, this represents a Gaussian mode (a spot) whereas other modes represent $LG_{lp}$ modes. For higher values of *l*, the beam will have phase fronts that look like intertwined spirals. Reprinted from ref. [55].

The expression for the electric field amplitude of a single LG mode is given by

$$LG_{lp}(r,\varphi,z) = C_{pl}\frac{w_0}{z_R w(z)}\left[\sqrt{2}\left(\frac{r}{w(z)}\right)\right]^{|l|} \times L_p^{|l|}\left(2\frac{r^2}{w(z)^2}\right)e^{-il\varphi} \times e^{-i(2p+|l|+1)\tan^{-1}\left(\frac{z_R}{z}\right)} \times e^{\left[\frac{-ikr^2}{2z\left(1+\left(\frac{z_R}{z}\right)^2\right)}\right]} \times e^{\left(-\frac{r^2}{w(z)^2}\right)} \quad (4)$$

The coefficient $C_{pl} = \sqrt[A]{\frac{p!}{(p+|l|)!}}$ is obtained at the requirement that every mode transmits the same amount of power. $L_p^{|l|}(x)$ is the generalized Laguerre polynomial given by

$L_p^{|l|}(x) = (-1)^{|l|} \frac{d^{|l|}}{d(x)^{|l|}} L_{p+|l|}(x)$. Where $p$ is the number of radial nodes in the intensity distribution, $l$ is the azimuthal mode number/index giving an OAM of $l\hbar$ per photon, $w(z)$ is the beam radius, $k$ is the free space wave number, $z_R$ signifies the Rayleigh range. In terms of their intensity cross-section, an LG mode with $l > 0$ comprises $p + 1$ concentric rings with a zero intensity on the axis [56]. Sjöholm [57] stated that the reason for using the absolute value of $l$ (that is, $|l|$) in the Laguerre polynomial and the amplitude part as seen in Equation (2) is that the Laguerre polynomials are only defined for $l > -1$ and that having the amplitude to the power of $-l$ would result in infinite amplitude as $r$ tends to 0.

## 3. Generation of OAM Beams

OAM or vortex beams in the optical domain can be generated in free space using spatial-generating devices and in optical fibers using fiber-generating devices.

### 3.1. Spatial-Generating Methods

OAM is usually generated by manipulating the spatial phase structure of the electric field of an electromagnetic wave [58]. A spatial device with a specially designed surface structure is used to imprint a phase profile on the electric field of a propagating plane light wave as it passes through the device. The resulting beam has a spiral or helical phased structure because of the spatial variation in the phase. A structured light wave can also be generated when an external magnetic field interacts with the light wave on a carbon nanotube [59]. When a magnetic field is applied to an optical device made from magneto-optical materials and a light wave is passed through, the refractive index of the wave becomes altered and this also changes the spatial phase distribution of the light [60,61].

The spatial-generating methods include the use of spatial devices such as cylindrical lenses [62–66], spiral phase plates [67–74], Phase holograms [75–81], spatial light modulators [82–88], and q-plates [89–92].

Allen's experiment established high-order Hermite-Gaussian (HG) modes are transformed into a helically phased beam with an azimuthal dependence $\exp(-il\varphi)$, a phase singularity or vortex at the center. The conversion process is performed in various optical materials known as mode converters shown in Figure 5. A fundamental transverse mode (a Gaussian beam, $TEM_{00}$) emitted from a laser, makes a small angle as it passes through the optical medium. This gives a twist to the resulting beam called the Laguerre-Gaussian beam [93].

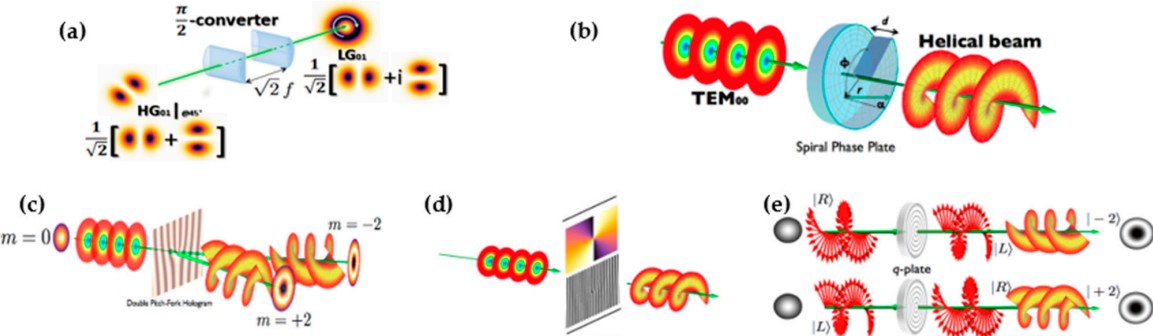

**Figure 5.** Exemplification of OAM spatial generation techniques: (**a**) Cylindrical lens which was used by Allen et al. to transform HG modes to LG modes with OAM and a phase singularity at the center; (**b**) spiral phase plate is the simplest converter. It has spiral phase distribution which converts a plane Gaussian beam of a particular wavelength into an OAM beam that has a spiral wavefront; (**c**) phase hologram with the use of forked diffraction grating, an OAM beam is formed as it passes through the forked grating. The number of forks determines the OAM number of the beam generated; (**d**) spatial light modulator uses the molecules in their liquid crystal to deflect and give a twist to the incident plane beam; (**e**) Q-plates are inhomogeneous birefringent devices that convert Gaussian beams to OAM beams.

It is important to note that distance and coherence length of propagation have significant effects on the propagation and generation of a vortex beam. As the vortices propagate farther or as the coherence length decreases, a vortex of an opposite sign is formed, resulting in two vortices in space; however, this does not affect the properties of the vortex beam, making it potentially useful in optical communications [94].

It should also be noted that these methods are plagued with disadvantages such as large volume space requirement by the spatial devices, high insertion loss due to the high refractive index difference at the interphase where the incident plane wave is being converted to OAM beams, a limited number of OAM beams created per time, and a high cost of fabrication of the optical materials as new materials need to be fabricated for the desired OAM number. Table 2 features a comparison of the different spatial OAM generation methods.

**Table 2.** Comparison of the spatial generation techniques of OAM.

| Spatial Generation Methods | Cylindrical Lens | Spiral Phase Plate | Phase Hologram | Spatial Light Modulator | Q-Plates | Metamaterials |
|---|---|---|---|---|---|---|
| Cost | Normal | low | Low | High | high | Low |
| OAM modes generated | Single | Single | Single | Single/Multiple | Single | Single |
| Flexibility | Low | Low | Low | High | High | Low |
| Transmission distance | Short | Short | Short | Short | Short | Short |
| Is it passive? | Yes | Yes | Yes | Yes | Yes | Yes |
| Can it withstand high power? | Yes | Yes | No | No | Yes | No |
| Processing difficulty | Low | Low | High | High | Low | High |
| Does it enable space division multiplexing? | No | No | No | No | No | No |

Still in the spatial domain, generating OAM beams at the nanoscale is of significant interest for various applications in nanophotonics [95,96] and optical device miniaturization [97]. Recent developments in the creation and detection of optical vortex as well as some of its applications are observed in photonic integrated circuits [98–103], micro-ring resonators [104–111], metamaterials [112–120], plasmonic nanostructures [121–124], and metasurfaces [125–128], which offer precise control over the phase, amplitude, and polarization of light at the nanoscale. By engineering the meta-atoms on a metasurface, it is possible to design them to introduce a tailored phase gradient, resulting in the generation of OAM beams. Metasurfaces can be fabricated with various materials, including plasmonic nanostructures or dielectric resonators, allowing for efficient manipulation of light and generation of OAM beams at the nanoscale.

In general, at the time of writing this review, it suffices to say that the spatial-generating methods have many limitations in the fabrication of the devices and cost among others, and cannot be commercialized profitably; hence, other methods need to be embraced.

### 3.2. Fiber-Generating Methods

The second method of generating OAM modes is the so-called fiber-generating method. This is more advantageous owing to characteristics such as miniaturization, lower insertion loss in fibers, increased transmission distance, higher efficiency, and a reduction in external interference, which is lacking in the spatial generating method [129]. Moreover, owing to the cylindrical shape of fibers, the eigenmode of the incident wave is easily restricted to the cylindrical symmetry of the fiber during conversion into OAM beams.

Conventional optical fibers cannot transmit OAM; hence, new specialized fibers need to be designed to support OAM mode transmission. These specialized fibers, however, require converters listed below and shown in the experimental setups in Figure 6.

(a)    Fiber gratings [130] used in FMF [131];
(b)    Mode-selective couplers [132] used in both SMF and FMF;
(c)    Photonic lanterns [133,134] also used in SMF;
(d)    Microstructure optical fibers or photonic crystal fibers (PCF) [135–137].

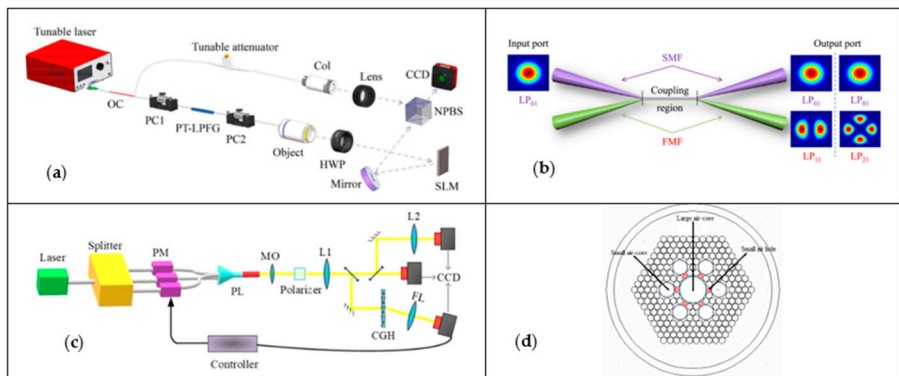

**Figure 6.** Experimental setup of OAM generation through fibers. (**a**) Fiber grating: Experimental setup to generate and detect 4th-order OAM modes by presetting the number of twists before the fabrication of the fiber grating. OC-optical coupler; PC-polarization controller; PT-LPFG—preset twist-long period fiber gratings; HWP-half-wave plate; SLM—spatial light modulator; Col—collimator; NPBS—non-polarization beam splitter. Reprinted from ref. [138]; (**b**) mode selective couplers which uses a single-mode fiber (SMF) as the input fiber and FMF as the output fiber joined together with fiber optic couplers. The mode coupler ensures that the refractive index of the fundamental Gaussian at the input fiber is retained in the resultant OAM mode at the output fiber. Reprinted from ref. [139]; (**c**) photonic lanterns: The fundamental mode is emitted from the laser passes through the splitter into the photonic lantern which couples and converts the beam into the desired OAM beam. PM—phase modulators; PL- photonic lantern; CGH—computer-generated hologram; MO—microscope objectives; L1, L2—imaging lenses; FL—Fourier lens. Reprinted from ref. [140]; (**d**) Photonic crystal fiber. Reprinted from ref. [141].

*3.3. Photonic Crystal Fibers (PCF)*

Although fibers that support OAM states have had limited success because they have only been able to propagate very short distances and the modal intensity patterns are unstable due to the mode mixing of the eigenmodes [142,143], specially designed microstructure optical fibers (MOFs), also known as photonic crystal fibers (PCF), have shown incredible abilities with flexible and adjustable fiber parameters structure [144]. Most especially, structures comprising air holes [145–148]. These perform excellently and are easy to manufacture. Some PCF designs with air holes are as follows:

- A circular PCF (C-PCF) supporting 26 OAM modes [149];
- A C-PCF with square air holes in the cladding that supports 46 OAM [150];
- A unique PCF with square and circular air holes (SC-PCF) that support 86 OAM modes [151];
- A pure-silica-based PCF with a central round air hole that supports 114 OAM modes was also designed showing higher effective refractive index difference, lower confinement loss, and non-linear coefficient [152];
- A PCF with an ssk2 dense crown glass ring with optimized central air hole radius and annular region thickness stably transmitted 394 OAM modes [153];
- A total of 84 OAM modes [154], 110 OAM modes [155], and 166 OAM modes [156].

The several results generated through the finite element analysis of the above-listed PCFs show a result of an effective modal index, low/flat dispersion, small and controllable non-linearity, high birefringent, high mode quality, high performance, and avoid mode coupling. These qualities are needed for the best performance in long-distance communication

systems and hold a huge promise in high-performance optical communication systems. However, more improvements are seen with PCF designs made with low refractive index rings owing to their easier fabrication/manufacturing [157]. Table 3 highlights the advantages and disadvantages of each fiber-generating method, whereas Table 4 gives a general merit and demerit of the spatial method and fiber method according to their corresponding references in the texts.

**Table 3.** Advantages and disadvantages of the fiber-generation technique of OAM.

| Fiber-Generating Method | Advantages | Disadvantages |
|---|---|---|
| Fiber gratings | Compatibility with existing fiber structure; highly stable and robust, low loss | Limited OAM mode selection due to design and fabrication limitations; reduced mode purity in high-order modes; limited bandwidth |
| Mode selective couplers | High mode purity; design flexibility to generate different OAM modes for various applications; wide bandwidth | Complexity in fabrication, leading to high cost and limited scalability; sensitive to misalignment |
| Photonic lanterns | Compatibility with existing fiber structure; efficient mode conversion; mode flexibility | Complexity in fabrication, leading to high cost and limited scalability; coupling losses can cause reduced OAM purity |
| Photonic crystal fibers (PCF) | High mode purity; supports multiple modes across a wide bandwidth; compact | Complex design and fabrication; high losses due to complex waveguide design |

**Table 4.** Advantages and disadvantages of the fiber-generation technique of OAM.

| Method of Generation | Advantages | Disadvantages | Application |
|---|---|---|---|
| Spatial-generation methods | Beams can be shaped and manipulated with great versatility; advanced beam steering capabilities and control over beam direction; a wide range of applications | Vulnerability to environmental influences such as atmospheric turbulence, scattering, and absorption results in beam variations, effectiveness, and distortions. Alignment challenges between transmitting and receiving systems | Free-space optical communication; imaging; sensing; quantum information processing; interferometry; micromanipulations |
| Fiber-generation methods | Compatible with the existing fiber optic communication networks; robust and stable with minimal beam distortions; high data capability due to scalability in transmitting multiple OAM beams | Complexity in design and fabrication can result in high cost; inefficient coupling impacts mode purity and results in transmission loss; high modal crosstalk | Optical fiber communications; optical imaging and sensing; fiber laser |

### 3.4. Measurement of OAM Modes

As much as it is important to generate OAM beams, it is equally important to be able to measure and determine the number of modes an OAM beam possesses for use in diverse applications. The topological charge *l* or OAM mode of a helical or vortex beam can be measured by examining the interference pattern between the beam and a reference beam [158] and the use of a Mach–Zehnder interferometer for higher-order topological charges [159,160]. The OAM of a single photon can also be measured using a computer-generated hologram setup [161]; the fork dislocation of the hologram is used to generate the azimuthal phase dependence of $e^{il\varphi}$ on the incident beam, giving it a helical phase front. To turn the twisted beam back into a plane wave, a reverse hologram is used to focus the helical beam through a pinhole, and it is then detected as a plane phase front. Measurements of the OAM state of a beam with SLM-based annular gratings were also reported in [162] where the number of black fringes and the direction of the diffraction patterns indicate the size and sign of topological charge values, respectively.

These measurements can be completed theoretically and experimentally in close interaction [163]. Theoretical methods involve mathematical simulations and models to understand, predict, and optimize the design parameters for the generation of OAM in light beams. Experimental methods involve physically manipulating light beams with the use of various optical components mentioned in Section 3 to imprint a desired spatial phase structure onto the light beam, resulting in the creation of OAM modes. Theoretical models guide the development of experimental techniques by providing insights into the fundamental principles of OAM generation whereas experimental results inform and validate theoretical models by providing empirical data and feedback for refining theoretical understanding. However, both approaches adhere to the azimuthal symmetry and conservation of angular momentum associated with OAM. Both theoretical and experimental methods essentially contribute to advancing knowledge and applications of OAM in various fields, such as optical communications, imaging, and quantum information processing.

## 4. Multiplexing of Information-Carrying OAM Beams

When a Gaussian beam encoded with information passes through an optical element such as a spiral phase plate, it acquires an azimuthal phase dependence of $e^{il\varphi}$ and becomes an OAM beam with a helical phase front. Multiple of such OAM beams ($l = 1, 2, 3, \ldots, N$), each carrying its own information, can be multiplexed. Each helical beam develops a phase singularity (optical vortex with a bright ring and a dark center) while the wavefront spirals around the optical axis as they propagate. The orbital angular momentum of each OAM beam remains unchanged by its propagation through free space and spherical lenses [164].

The beam with the highest OAM number $N$ creates the outermost vortex by wrapping the smaller vortices around itself, whereas the helix with the lowest OAM number is located in the center of the multiplexed vortex structure. As a result, many OAM beams propagate as though they were a single beam.

After their transit through the communication channel, the OAM beam must be converted back to a Gaussian to extract the information contained in them. An inverse spatial element (SPP) with a predetermined charge of $-l$ is utilized and used for demultiplexing, isolating, and recovering each data-carrying OAM beam at the receiver end to eliminate the azimuthal phase dependence $e^{il\varphi}$ of the OAM beam and its information [165].

Most OAM beam multiplexing was experimentally demonstrated across relatively short distances of less than 1m in the lab often; however, in recent years, several investigations have investigated the possibility of employing OAM beams to create long-distance FSO linkages in the field environment [166,167].

### 4.1. OAM in Space-Division Multiplexing

Space-division multiplexing is a technology used in optical communication systems to boost capacity and data transmission capabilities by employing various spatial modes within an optical fiber or waveguide. Unlike TDM or WDM, which used time or frequency, SDM exploits the spatial domain to simultaneously transmit many independent data streams. Using fibers as the communication links with the SDM multiplexing techniques, reports show that 2Pbps/fiber was achieved [168–172], half of which has been unrealized with the other multiplexing methods. This is because each spatial path or mod4e in SDM is a separate/independent communication channel; hence, it is extremely useful in improving the capacity of communication systems.

SDM relies on multiplexing spatially separated beam modes such as linearly polarized (LP) modes [173–175] and OAM modes [176–178]. This is made possible by orthogonality, transverse phase structure, and a helical/twisted phase front of such modes. OAM multiplexing corresponds with mode-division multiplexing (MDM), which is a popular form of SDM multiplexing. MDM can increase the transmission capacity in optical fiber communications due to the orthogonality and theoretically infinite states of OAM modes [179] used in such a technique.

Based on their orthogonality, OAM beams could also be multiplexed with the other multiplexing techniques and transmitted over free space or fiber to increase the data capacity of communication systems by the number of beams transmitted. Likewise, the spectral efficiency is increased because the transmitted beams are in the same frequency range [180]. The first time OAM multiplexing was used for MDM communication links, four different OAM modes were multiplexed on two polarizations in free space in Figure 7a–c. This yielded a data rate of 1.37 Tbit/s [181].

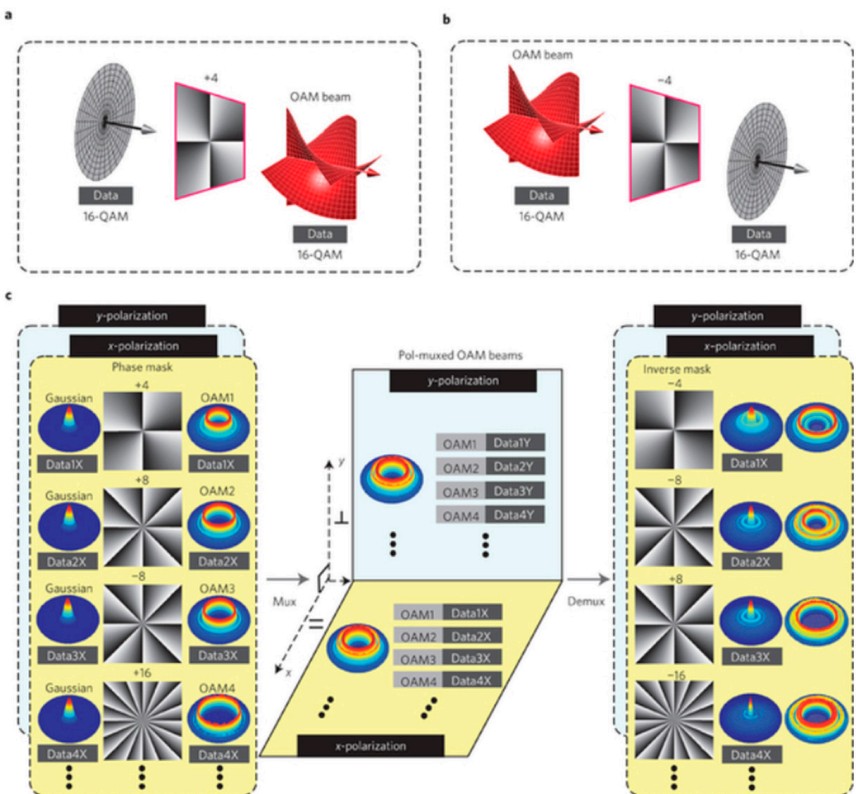

**Figure 7.** OAM multiplexing with PDM in free space. (**a**) Represents the generation of an information-carrying OAM beam after an information-carrying Gaussian has been passed through a spatial element (spiral phase mask) with OAM mode $l = 4$; (**b**) signifies the recovery of an information-carrying Gaussian from an information-carrying OAM beam after passing through an inverse spiral phase mask of $l = -4$; (**c**) describes the multiplexing and demultiplexing of the polarization multiplexed OAM beams. Reprinted from ref. [181].

In addition to PDM multiplexing, OAM beams can also be multiplexed with WDM to boost the capacity of OAM multiplexed in free space links [182]. In the experiment illustrated in Figure 8a,b, the OAM multiplexed beams and OAM-PDM multiplexed beams, respectively, are carried on the same frequency. Figure 8c depicts how other independent data channels can be sent or received using the other frequencies on the same number of OAM modes $N$ and both polarizations. Because OAM multiplexing and PDM are compatible with WDM in this manner, the aggregated connection capacity is boosted further by M times by using M separate carrier frequencies. Overall, the data rate of the communication link is $2 \times N \times M$, indicating that OAM is truly an additional degree of freedom to the PDM and WDM techniques. In the experimental setup, a data rate of 100Tbit/s was achieved with 12 OAM modes, two polarizations, and 42 wavelengths.

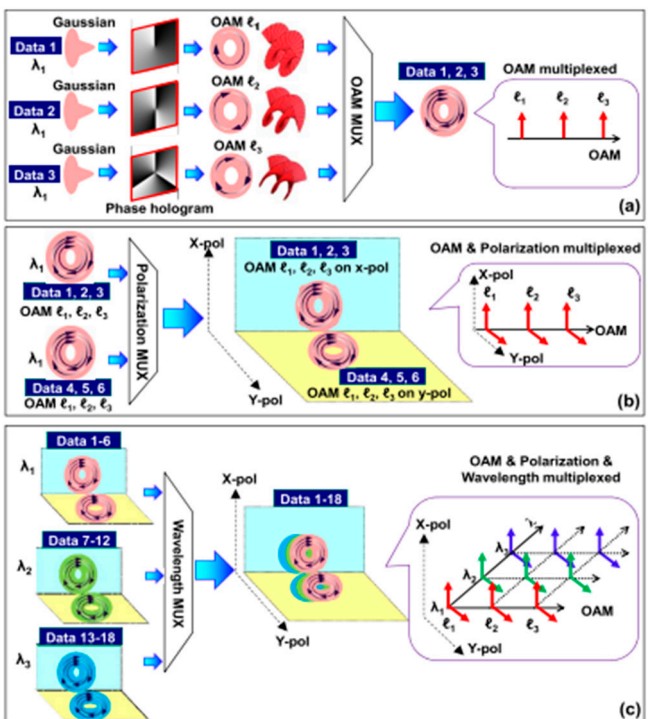

**Figure 8.** The idea of boosting channel capacity by combining OAM multiplexing with PDM and WDM. (**a**) Using OAM beams to multiplex numerous data channels; (**b**) combination of PDM and OAM multiplexing; (**c**) a total of 12 OAM modes, 2 polarizations, and 42 wavelengths were multiplexed to reach a 100.8 Tbit/s aggregated data rate by combining OAM multiplexing and PDM with WDM [182].

Aside from OAM modes, the SDM technique can also be used in multi-core fiber (MCF) [183], a fiber that consists of multiple individual cores within a single fiber. The number of cores in the fiber gives the fiber capacity. Moreover, few-mode fiber (FMF) [184], which supports multiple spatial modes within a single core, is employed in SDM. Here, the number of modes supported by the fiber gives the fiber capacity. SDM transmission allows a combination of multiple cores within a single cladding (MCF) or multiple modes in a single core (FMF) [185]. SDM thus positions itself as the multiplexing approach best suited to carry future capacity needs.

The advantages of OAM-based MDM systems are numerous. Apart from increasing the channel capacity and spectral efficiency of communication links due to the several spatial modes accommodated, SDM also provides scalability by enabling the addition of more spatial pathways or modes when the need for more capacity rises, without necessitating substantial changes to the underlying infrastructure. OAM-SDM also simultaneously enables significant reductions in cost-per-bit and increased energy efficiency [186]. Likewise, it will avoid complex multiple-input and multiple-output (MIMO) digital signal processing and overcome intermodal crosstalk and finite modes [187–189]. In the multiple-in multiple-out (MIMO) technique, the channel capacity increases in correspondence with the number of transmitting and receiving antennas [190,191], making it a bulky and expensive technique.

Although there are many advantages, the actual application of OAM-based multiplexing is still constrained significantly by crosstalk. Crosstalk between OAM modes can cause interference, poor signal quality, and impose a limit to the number of multiplexable channels [192,193].

### 4.2. OAM-Compatible Infrastructures and Devices for Scaling

Although OAM-SDM systems are taking the top research attention, Winzer [194] looked at the integration of SDM components into the existing non-SDM infrastructure. Data network designers believe that in order to use the currently embedded fiber struc-

tures to scale up the channel capacity and spectral efficiency of communication system, the OAM optical communications research community must come up with compatible new technologies such as devices such as photonics integrated circuits, light sources, (de)modulators, amplifiers, switching devices, OAM filters [195], transmitters, receivers, and signal processing elements which support compact, efficient, cost-effective OAM generation, multiplexing, demultiplexing, and detection [196,197]. Figure 9 shows some of the devices needed to fully scale the OAM-SDM technique.

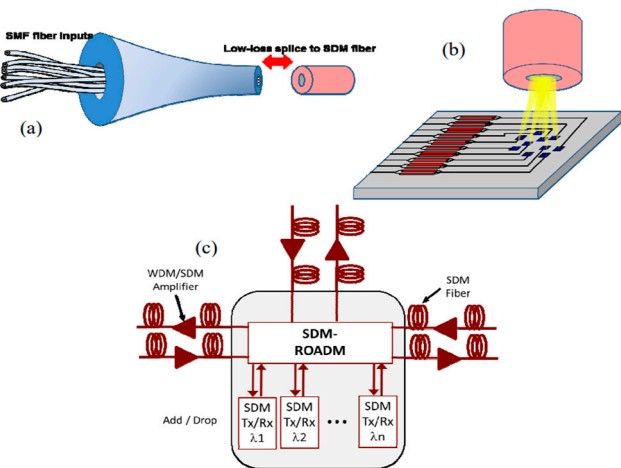

**Figure 9.** SDM-compatible devices. (**a**) Elegantly scalable passive multiplexers, (**b**) integrated transmitter and receiver arrays providing low-loss coupling to many modes of an SDM fiber, and (**c**) reconfigurable routing elements that can direct SDM traffic without the need for electronic MIMO in between transmitter and receiver. Reprinted from ref. [198].

In terms of commercialization of the SDM multiplexing technique, an optical lab named "Cailabs" has developed a solution using multi-plane light conversion (MPLC) technology to generate high-order, high-purity OAM beams in free space and fiber [199]. The MPLC technology converts multiplexed modes from SMF inputs by using successive reflections on a single reflective phase plate and a mirror. The technology then converts each input into a given mode within an MMF or free space output. Since each mode represents an additional information channel, this means that there would be an increase in the bandwidth which is the goal of all OAM research. The setup also consists of a demultiplexer which recovers the original signal information. According to the company, their solution overcomes all the other constraints by the other generation techniques featuring a high number of modes (up to 45 modes), optimal crosstalk, low insertion loss level, completely passive, and compatibility with a wide range of wavelengths. It is the only commercial solution that offers OAM-SDM multiplexing at the time of writing this review.

Several commercial enterprises also demonstrated wireless transmission for multiplexed OAM beams [200–202], achieving a data rate $100\times$ that of LTE and Wi-Fi. These can be used for next-generation 5G and 6G systems, such as connected cars, virtual reality/augmented reality (VR/AR), high-definition video transmission, and remote medicine.

## 5. Summary and Outlook

### 5.1. Summary

This work offers an overview of the orbital angular momentum components of electromagnetic waves and their capabilities in boosting the optical communication channel to meet the growing capacity needs of future technologies. Firstly, the authors reviewed why OAM beams are important, especially in optical communications. The background of OAM was presented and the various methods of generating OAM in space, such as the use of spiral phase plate, cylindrical lens, and others that convert incident Gaussian beams to beams with twisted wavefronts also known as Laguerre-Gaussian mode. The orthogonality

of these modes makes them suitable for various applications and makes it possible for them to be multiplexed with the other multiplexing techniques to generate higher channel capacity. Additionally, the authors reviewed how OAM beams are being generated in specialized optical fibers with the use of fiber gratings, couplers, and especially the photonic crystal fibers that support OAM mode transmission needed in high-performance optical communication. Contexts of multiplexing OAM with SDM, PDM, and WDM were also considered.

*5.2. Outlook and Future Perspectives*

There are still many questions and challenges to be overcome in OAM research. These challenges can be turned into opportunities for the future development and utilization of OAM. It would be interesting to see more research towards the design of efficient cost-effective and compatible multiplexing and demultiplexing devices that would support a large number of OAM-SDM modes [203] including some advances in tunable, compact, and high-power OAM lasers [204–207]. Multiplexing of OAM and WDM was recently experimented with in a multimode fiber to also improve capacity [208], so this could also be considered to keep pushing the limits and scope of OAM beams. In future research, the development of cutting-edge fibers which are fit for SDM and which will also be able to accommodate the next frontier technologies needs to be investigated. Crosstalk between channels is a clear potential drawback that is continuously being addressed [209,210]. An interdisciplinary exploration of OAM in other scientific disciplines can also lead to new discoveries and applications in diverse fields.

With ongoing attempts to investigate and enhance various methodologies, including multicore fibers, few-mode fibers, and OAM-based systems, SDM remains a topic of current study and development. These developments in SDM will help optical communication systems fulfill the goal of introducing OAM as an additional degree of freedom and increasing optical channel capacities. At the current rate of research in the field, it is believed that this feat would be achieved in the near future and commercialization would commence on a huge scale.

**Author Contributions:** Conceptualization, T.M.O. and P.A.R.; writing—original draft preparation, T.M.O.; writing—review and editing, T.M.O., P.A.R. and M.R.; supervision, P.A.R. and M.R.; funding acquisition, P.A.R. and M.R. All authors have read and agreed to the published version of the manuscript.

**Funding:** This research was funded by the Portuguese National Funding Agency (FCT-MCTES), UIDB/04559/2020 (LIBPhys), and UIDP/04559/2020 (LIBPhys).

**Institutional Review Board Statement:** Not applicable.

**Informed Consent Statement:** Not applicable.

**Data Availability Statement:** Not applicable.

**Conflicts of Interest:** The authors declare no conflict of interest.

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
