# Peer review of "Generation of Photon Orbital Angular Momentum and Its Application in Space Division Multiplexing"

_photonics, doi:10.3390/photonics10060664_

Round 1

Reviewer 1 Report

This work offers an overview of the orbital angular momentum components of electromagnetic waves and their capabilities in boosting the optical communication channel to meet the growing capacity needs of future technologies. Firstly, the authors reviewed why OAM beams are important, especially in optical communications. The background of OAM was presented and the various methods of generating OAM in space, such as the use of spiral phase plate, cylindrical lens and others which convert incident Gaussian beams to beams with twisted wavefronts also known as Laguerre Gaussian mode. The orthogonality of these modes makes them suitable for various applications and makes it possible for them to be multiplexed with the other multiplexing techniques to generate higher channel capacity. Additionally, the authors reviewed how OAM beams are being generated in specialized optical fibers with the use of fiber gratings, couplers, especially the photonic crystal fibers that support OAM mode transmission needed in high-performance optical communication.

the manuscript is rather well written and it sounds useful for a generic reader interested in the topic. 

I don't find particular criticisms, anyway, the authors completelly missed to mention an important approach for the generation, multiplexing and control of OAM beams in particular at the nanoscale where optical devices can find important applications. For example, some interesting works on this topic are:

doi.org/10.1038/s41377-021-00668-6

doi.org/10.1039/C5NR07374J

doi.org/10.1039/C7NR00124J

doi.org/10.1364/OL.39.004899

doi.org/10.1021/acsphotonics.2c01321

  •  

the english grammar is almost ok

Author Response

Reviewer #1:

Point 1: This work offers an overview of the orbital angular momentum components of electromagnetic waves and their capabilities in boosting the optical communication channel to meet the growing capacity needs of future technologies. Firstly, the authors reviewed why OAM beams are important, especially in optical communications. The background of OAM was presented and the various methods of generating OAM in space, such as the use of spiral phase plate, cylindrical lens, and others that convert incident Gaussian beams to beams with twisted wavefronts also known as Laguerre Gaussian mode. The orthogonality of these modes makes them suitable for various applications and makes it possible for them to be multiplexed with the other multiplexing techniques to generate higher channel capacity. Additionally, the authors reviewed how OAM beams are being generated in specialized optical fibers with the use of fiber gratings, couplers, and especially the photonic crystal fibers that support OAM mode transmission needed in high-performance optical communication.

Answer from Authors: Thank you. The manuscript was improved in accordance with the suggestions. Page 18 of the revised manuscript.

Point 2: the manuscript is rather well written, and it sounds useful for a generic reader interested in the topic. 

Answer from Authors: Thank you for your comment and we are happy you found it so. Yes, the review is written for and targeted at readers who want to have a general knowledge of orbital angular momentum and its capabilities.

Point 3: I don't find particular criticisms, anyway, the authors completely missed to mention an important approach for the generation, multiplexing, and control of OAM beams in particular at the nanoscale where optical devices can find important applications. For example, some interesting works on this topic are:

doi.org/10.1038/s41377-021-00668-6

doi.org/10.1039/C5NR07374J

doi.org/10.1039/C7NR00124J

doi.org/10.1364/OL.39.004899

doi.org/10.1021/acsphotonics.2c01321

Answer from Authors: Thank you very much for the suggestion and article recommendations. A new paragraph has been included in section 3.1, pages 8-9 of the manuscript.

Once again, we appreciate the value your feedback adds to the review process. Your suggestions and knowledge certainly enhanced the quality of our work. To make sure the reader can comprehend the larger context of the revised manuscript, we have taken your feedback into consideration and made additional attempts to improve the clarity and organization of the document.

If you have any further suggestions or recommendations, please do not hesitate to let us know.

Thank you for your time and consideration.

Reviewer 2 Report

The manuscript ID photonics-2397150 mainly present an analysis related to particular techniques that can be involved in the generation of photon orbital angular momentum. Additionally, the authors discuss about some applications in space division multiplexing based on the photon orbital momentum studied. Please see below a list of comments to the authors:

1. It is suggested to include within the introduction a graphical abstract describing the aim of the work together to the characteristics studied in this review in order to easily visualize the goals of the study undertaken in this manuscript.

2. Figure 6 is useful; but a roadmap of the topic is missing in the report.

3. The authors should clearly state in the introduction what this review adds to literature in respect to the publications presented and other reviews in comparative topics. You can see for instance https://doi.org/10.3389/fphy.2021.773505

4. The photon orbital angular momentum can be obtained by different physical effects and not only purely optical. The authors are invited to see for instance the potential participation of a magnetic field, you can see for instance: DOI: 10.1039/d1cp05195d

5.  Advantages and disadvantages of the different techniques for generation of a photon orbital angular momentum must me confronted.

6. It would be useful if the authors can comment how can be the differences or symmetry in experimental and theoretical generation of photon orbital angular momentum.

7. Section 4 and 5 should be extended and described with better details. A deeper analysis ought to be expected for these sections regarding that they are the main topic of the work.

8. Future perspectives for the generation of photon orbital angular momentum and multiplexing applications must be described.

9. The authors could consider update some references to the date of the review stage.

10. The presentation of the collective form for citations should be split in order to better justify and present by using individual expressions each reference selected to be part of this review.

A proofreading is mandatory. Section 5 is currently numbered as section 3.

Author Response

Reviewer #2:

The manuscript ID photonics-2397150 mainly presents an analysis related to techniques that can be involved in the generation of photon orbital angular momentum. Additionally, the authors discuss about some applications in space division multiplexing based on the photon orbital momentum studied. Please see below a list of comments to the authors:

Point1. It is suggested to include within the introduction a graphical abstract describing the aim of the work together to the characteristics studied in this review in order to easily visualize the goals of the study undertaken in this manuscript.

Answer from Authors: Thank you for your great suggestions. A graphical abstract has been added to depict the goal of the review.

Point2. Figure 6 is useful, but a roadmap of the topic is missing in the report.

Answer from Authors: Yes, and in fact, the figure has now been moved to the introduction section of the revised manuscript and renamed as Figure 1 to better explain the goal of the study and for the readers to quickly see why SDM techniques are being considered widely in optical communication research.

Regarding the issue of the missing roadmap in the report, truly, we acknowledge this as well as it is important to provide the readers with a clear overview and structure of the subject matter. As a result, we have added a new subsection titled “Roadmap of the review” on page 4, just below the introduction. We provided a concise yet comprehensive roadmap that outlines the main components of the review for the readers to navigate the sections easily.

Point3. The authors should clearly state in the introduction what this review adds to the literature in respect to the publications presented and other reviews in comparative topics. You can see for instance https://doi.org/10.3389/fphy.2021.773505

Answer from Authors: Following your second comment, we have also revised the introduction for a better grasp of the main ideas of the review. We stated what the review adds to the literature in the last paragraph of the introduction on page 4.

Point4. The photon orbital angular momentum can be obtained by different physical effects and not only purely optical. The authors are invited to see for instance the potential participation of a magnetic field, you can see for instance: DOI: 10.1039/d1cp05195d

Answer from Authors: Thank you for pointing it out and for the reference. A paragraph was added on page 8 in section 3.1 to address this.

Point5.  The advantages and disadvantages of the different techniques for the generation of a photon orbital angular momentum must be confronted.

Answer from Authors:  Tables were introduced to show comparisons, advantages, and disadvantages of OAM generation techniques in section 3, pages 9, 11, and 12.

Point6. It would be useful if the authors can comment how can be the differences or symmetry in the experimental and theoretical generation of photon orbital angular momentum.

Answer from Authors:  Thank you for this comment. As a result, we realized we needed to add more context to the review. Section 3.3 titled “Measurement of OAM Modes” was added to describe the symmetries.

Point7. Sections 4 and 5 should be extended and described with better details. A deeper analysis ought to be expected for these sections regarding that they are the main topic of the work.

Answer from Authors:  True, your comments made this section of the entire review better. We realized that the content in section 4 of the submitted manuscript was well suited to be in the introductory section so that the readers can grab the concept of multiplexing earlier, being one of the major ideas of the review before they dive deeper into the more complex aspects. More details about multiplexing were included on pages 13 to 17.

Point8. Future perspectives for the generation of photon orbital angular momentum and multiplexing applications must be described.

Answer from Authors:  A new subsection (5.2) titled” Outlook and Perspectives” was added on page 18.

Point9. The authors could consider updating some references to the date of the review stage.

Answer from Authors: new references up to April 2023 have been included.

Point10. The presentation of the collective form for citations should be split in order to better justify and present by using individual expressions for each reference selected to be part of this review.

Answer from Authors:  Thank you for the valuable feedback on our manuscript. We appreciate your suggestion to split the citations and present individual expressions for the selected references. However, for this review, our goal was to offer a collective overview of the references which we have carefully chosen based on their relevance to the discussion. Due to this, it would be challenging to provide extensive justifications for each reference without compromising the scope and goal of presenting a broad understanding of the literature concisely.

Once again, we appreciate the value your feedback adds to the review process. Your suggestions and knowledge certainly enhanced the quality of our work. To make sure the reader can comprehend the larger context of the revised manuscript, we have taken your feedback into consideration and made additional attempts to improve the clarity and organization of the document.

If you have any further suggestions or recommendations, please do not hesitate to let us know.

Thank you for your time and consideration.

Round 2

Reviewer 2 Report

In my opinion, the authors have successfully address most of the points raised in the review stage. The analysis is clear and it can be useful for future research related to generation of photon orbital angular momentum and its application in space division multiplexing. Then I can recommend this work for publication in present form.

Some expressions can be improved.